# Burnout in Pediatric Oncology: Team Building and Clay Therapy as a Strategy to Improve Emotional Climate and Group Dynamics in a Nursing Staff

**DOI:** 10.3390/cancers17071099

**Published:** 2025-03-25

**Authors:** Antonella Guido, Laura Peruzzi, Matilde Tibuzzi, Serena Sannino, Lucia Dario, Giulia Petruccini, Caterina Stella, Anna Maria Viteritti, Antonella Becciu, Francesca Bianchini, Deborah Cucculelli, Carmela Di Lauro, Ivana Paglialonga, Sabina Pianezzi, Roberta Pistilli, Sabrina Russo, Paola Adamo, Daniela Pia Rosaria Chieffo, Dario Talloa, Alberto Romano, Antonio Ruggiero

**Affiliations:** 1UOS di Psicologia Clinica, Fondazione Policlinico Universitario Agostino Gemelli IRCCS, 00168 Rome, Italy; antonella.guido@guest.policlinicogemelli.it (A.G.); laura.peruzzi@guest.policlinicogemelli.it (L.P.); serenasannino1994@gmail.com (S.S.); danielapiarosaria.chieffo@policlinicogemelli.it (D.P.R.C.); 2UOC di Oncologia Pediatrica, Dipartimento della Salute della Donna, del Bambino e di Sanità Pubblica, Fondazione Policlinico Universitario Agostino Gemelli IRCCS, 00168 Rome, Italy; lucia.dario@policlinicogemelli.it (L.D.); giulia.petruccini@policlinicogemelli.it (G.P.); caterina.stella@policlinicogemelli.it (C.S.); annamaria.viteritti@policlinicogemelli.it (A.M.V.); antonella.becciu@policlinicogemelli.it (A.B.); francesca.bianchini@policlinicogemelli.it (F.B.); deborah.cucculelli@policlinicogemelli.it (D.C.); carmela.dilauro@policlinicogemelli.it (C.D.L.); ivana.paglialonga@policlinicogemelli.it (I.P.); sabina.pianezzi@policlinicogemelli.it (S.P.); roberta.pistilli@policlinicogemelli.it (R.P.); sabrina.russo@policlinicogemelli.it (S.R.); dario.talloa@gmail.com (D.T.); alberto.romano@guest.policlinicogemelli.it (A.R.); 3Fondazione Lene Thun, 39100 Bolzano, Italy; matilde-t@hotmail.com (M.T.); paola.adamo@thun.it (P.A.); 4Department of Woman and Child Health and Public Health, Università Cattolica del Sacro Cuore, 20123 Rome, Italy

**Keywords:** burnout, pediatric oncology, nurses

## Abstract

This study analyzes burnout among healthcare workers in a pediatric oncology unit. To prevent and mitigate the identified risk factors, a pilot project was conducted to enhance the emotional well-being of nursing staff at the pediatric oncology unit of the Fondazione Policlinico Universitario Agostino Gemelli IRCCS in Rome. The objective of the study was to evaluate the impact of a team-building course that incorporates art and clay therapy in promoting the well-being of healthcare workers and in reducing burnout levels. The results showed the high emotional burden of healthcare workers and the effectiveness of the pilot project in alleviating burnout. Our findings emphasize the need to implement innovative support programs that aim to promote a positive work environment while also fostering the well-being of the operators.

## 1. Introduction

Burnout represents an increasingly common phenomenon among healthcare professionals. It is characterized by conditions of physical, emotional, and mental exhaustion caused by chronic stress at work [1]. Healthcare workers working in particular settings, such as pediatric oncology, are exposed to significant levels of stress because of the complex and often traumatic nature of their work, which involves treating children with life-threatening illnesses. A systematic review and meta-analysis indicate that pediatric oncology nurses are especially vulnerable to emotional exhaustion and exhibit moderate levels of depression despite experiencing high personal accomplishment [2]. According to Pellegrino [3], burnout symptoms can manifest in various ways, including physical: insomnia, headaches, and general discomfort; behavioral: irritability, impulsivity, and isolation; and cognitive-affective: depression, hypersensitivity/insensitivity, cynicism, pessimism, and inattention.

Healthcare professionals in complex work environments, like pediatric oncology departments, engage in emotionally demanding relationships with patients. These relations between the caregiver and the patients can lead to significant psychological stress at both individual and group levels. Studies have identified several factors contributing to burnout risk, including the type of disease—cancer diagnoses are often perceived as “incurable” or “inescapable”—the emotional intensity of relationships with patients, the management of pain and suffering, and the inevitable and frequent exposure to death [4]. Burnout affects not only the mental and physical well-being of healthcare workers but also the quality of care provided and their relationship with patients and families. Therefore, it is essential to identify strategies to prevent and manage burnout in a complex setting such as pediatric oncology. The most recent scientific evidence defines art as a possible tool for preventing burnout among healthcare workers. Artistic practices in healthcare settings have been linked with stress reduction, improved emotional well-being, and enhanced resilience [5].

Among patients, art therapy facilitates emotional expression, personal reflection, and connection with self and others [6]. Similarly, for healthcare workers, it provides strategies for dealing with work-related traumatic experiences. Several studies have investigated the potential of art as a supportive tool for healthcare providers, highlighting how it can foster greater emotional awareness and a sense of personal gratification [7]. In pediatric oncology, art-based interventions could help create reflective and supportive spaces, promoting well-being in healthcare workers, improving stress management, and reducing burnout risk.

This article presents the Art-Out pilot project, a team-building course integrating clay therapy to reduce burnout among healthcare workers.

## 2. Materials and Methods

In 2022, the Art-Out pilot project aimed at the healthcare staff of pediatric oncology unit at the Fondazione Policlinico Universitario A. Gemelli IRCCS in Rome, using clay therapy to prevent burnout and improve the emotional climate of the team. The group was led by a ceramist from the Lene Thun Foundation and by a psychologist-psychotherapist from the pediatric oncology unit. The project duration was 10 months; the project culminated in the development of a work of art created by the group of participants and subsequently displayed in the hospital’s public spaces.

The primary objective of the study was to test how much the team-building course, through clay therapy, helped improve the work climate and promoted well-being in healthcare workers, reducing burnout levels. The structure of the multidisciplinary project included respective steps:‑Literature review to investigate the phenomenon, with focus on pediatric oncology-hematology areas;‑Assessment (T0) of the degree of nursing staff burnout, alexithymia, and emotional dysregulation by validated screening tests (MBI, TAS-20, DERS);‑Implementation of team-building through clay therapy with two monthly meetings for the nursing team (10 total);‑Final evaluation (T1) of how the variables under study changed.

### 2.1. Participants

The population involved in the study consists of the nursing team affiliated with the pediatric oncology unit at the Fondazione Policlinico Universitario A. Gemelli IRCCS. Each participant completed a questionnaire to facilitate the collection of demographic data. Initial recruitment took place in November 2022. Data collection was conducted pre- and post-intervention: from 16 to 31 November 2022 at T0 and from 15 to 31 September 2023 at T1. No financial incentives were provided to participants; instead, they demonstrated motivation and interest; the participants coordinated their attendance with their supervisor based on their shifts. Their participation was voluntary, and their response was considered informed consent for participating in the study. Complete confidentiality of data collected was maintained.

### 2.2. Tools

The following psychological tests were used to monitor the variables under examination: Maslach Burnout Inventory (MBI), Toronto Alexithymia Scale (TAS-20), and Difficulties in Emotion Regulation Scale (DERS).

The presence and degree of burnout among healthcare workers were assessed through the Italian version of the Maslach Burnout Inventory (MBI) [8]. This questionnaire has been widely used in literature regarding mental health in healthcare professionals and it’s the standard tool for measuring burnout among healthcare workers [9]. The instrument assesses an individual’s level of burnout by asking respondents to answer how often certain emotions are experienced over the course of a year. On a 7-point Likert scale, respondents rate how often they experience certain emotions, from 0 (never) to 6 (every day). The questions assess the three main subscales of work-related feelings: emotional exhaustion, depersonalization, and personal accomplishment. The emotional exhaustion subset consists of nine questions, the depersonalization subset includes five questions, and the personal fulfillment subset includes eight questions. A high level of burnout is defined as a score of 30 or higher in the emotional exhaustion subset, a score of 12 or higher in the depersonalization subset, or a score of 33 or lower in the personal achievement subset. The categories of burnout levels are “high”, “moderate”, and “low”. High levels of burnout are correlated with higher scores in emotional exhaustion and depersonalization items and lower scores in personal accomplishment.

In this analysis, alexithymia was assessed using the Italian version Toronto Alexithymia Scale (TAS-20) with twenty items [10]. Many studies have used the TAS-20 test to measure levels of alexithymia in healthcare workers [11,12,13,14,15,16,17,18]. The literature highlights how burnout can be closely linked to “emotional blindness”, a defense mechanism against negative and overwhelming emotions known as alexithymia [19,20].

The TAS-20 has a three-factor structure: factor 1 assesses the ability to identify feelings and distinguish between feelings and the bodily sensations of emotional arousal (difficulty in identifying feelings); factor 2 reflects the inability to communicate feelings to other people (difficulty in describing feelings); and factor 3 assesses outward-oriented thinking. A score of 61 or higher was considered indicative of alexithymia.

The presence of any difficulties in the regulation of negative emotions was assessed through the Difficulties in Emotion Regulation Scale (DERS), a questionnaire also used in the literature for healthcare professionals [21,22] that specifically investigates awareness and acceptance of one’s emotions and the ability to use appropriate emotional regulation strategies [23]. The original version was presented by Gratz and Roemer [11] and was later translated and adapted to Italian by Sighinolfi et al. [12]. The DERS scale is a 36-item measure designed to assess clinically relevant difficulties in emotion regulation. The scale provides valuable insights into an individual’s ability to understand, accept, and manage emotions effectively. It consists of 36 items with a 5-value Likert scale ranging from “almost never” to “almost always”. It includes 6 scales: non-acceptance of negative emotions (non-acceptance: 6 items), difficulties engaging in goal-directed behaviors when distressed (goals: 5 items), belief that there is little that one can do to regulate emotions effectively (strategies: 8 items), difficulties controlling impulsive behaviors when distressed (impulse: 6 items), lack of emotional awareness (awareness: 6 items), and lack of emotional clarity (clarity: 5 items) [11,12].

### 2.3. The Art-Out Project

The Art-Out is a pilot project that aims to improve emotional climate and prevent and limit burnout in healthcare personnel. Nursing staff working in pediatric oncology unit were involved in this project using art and its therapeutic value.

The Art-Out Project combines the experience of team building with clay therapy, promoting the creative process fostered by the manipulation of clay and its benefits. The nursing team participated in focus groups focusing on key themes that emerged during the two group brainstorming sessions (meetings 1 and 5). Each two-hour meeting consisted of a group work phase on the significant themes that emerged and a decompression phase, characterized by working with clay, which allowed for further exploration of each shared theme and left a trace of it through the expressive channel.

The meetings addressed various areas of vulnerability, including communication, individual and group boundaries, emotional responses to the work environment, and group dynamics that need monitoring like personal and interpersonal conflict, tension, cohesion, confusion, boredom, motivation, peer pressure, subgroup formation, and unclear goals. Each meeting fostered group sharing of emotions, experiences, and thoughts related to the professional context. Clay work, as a decompressive phase, provided a tangible representation of the issue discussed during the meeting.

The culminating artistic piece, created by the nursing team at the end of the 12 sessions, was entitled “The Burnout Spiral” (Appendix A). The artistic piece encapsulates the group’s journey, serving as a kind of shared “emotional signpost” born from abstract concepts such as fear, empathy, and active listening, into a figurative form. Each emotional insight or warning was modeled on soft clay tiles of equal size, which were then assembled into a permanent installation in the hospital lobby, serving as a visible trace in the hospital environment of the path undertaken together by the group. Additionally, the team-building experience allowed members to promote and improve communication within the nursing team, highlighting critical challenges like available resources and protective and risk factors within the work environment.

### 2.4. Statistical Analysis

In the absence of data on burnout within the target population, we set the sample size for this study at 13 subjects, which corresponds to the total number of nurses from our pediatric oncology unit who decided to participate in the pilot project. Despite the small sample size, it captures various expected proportions, with a 95% confidence level and a margin of error ranging from a minimum of 2% (for an expected proportion of 1%) to a maximum of 10.2 (for an expected proportion of 50%). Quantitative variables were presented through mean and standard deviation (SD), and qualitative variables were presented through absolute and percentage frequency tables. Comparison of means was performed by Mann–Whitney test, which was considered statistically significant for *p*-values < 0.05. Statistical evaluation was performed by XLSTAT 2023.1.4.1408.

## 3. Results

In this study of the 19 members of the nursing team, 13 (68.4%) took part in the project. The detailed demographic variables of the participants (Table 1) included age, gender, role in the department’s care team, professional seniority and length of service within the pediatric oncology unit, and the presence or absence of personal psychotherapy. The age range was 23 to 61 years, and only one man participated in the study. Each participant followed at least 8 (80%) of the 10 meetings scheduled for each group, and all of them completed the study assessments pre-intervention (T0) and at the conclusion of the project (post-intervention T1).

The prevalence study conducted on the enrolled nursing staff (N = 13) analyzed the burnout risk levels, alexithymia, and emotional dysregulation before (T0) and after (T1) the team-building course. In MBI at T0 (Table 2) compared with the baseline cut-off (Table 3), medium risk was found in personal accomplishment (T0 = 35.23), high risk was found in depersonalization (T0 = 12.07), and medium risk was found in emotional exhaustion (T0 = 19.07). At T1 in the MBI, all the variables analyzed improved from the baseline cut-off (Table 3). In particular, the variable “emotional exhaustion”, which had a value of 19.07 at T0 (medium risk), reduced statistically significantly at T1, reaching a value of 12.92 (*p* = 0.039), placing it below the clinical reference range. “Depersonalization” showed a statistically significant decrease at T1 (*p* = 0.013). Our results showed significant improvement in the MBI total score (Figure 1) (*p* = 0.019).

In TAS-20 (Table 4), the total score assessed at T0 (T0 = 61.15) falls within the clinically significant range for alexithymia (cut off: ≥61 Alexithymia; 52–60 Borderline; ≤51 No alexithymia). Ten months later, the score comparison performed at T1 (T1 = 50.15) showed a significant (*p* = 0.028) improvement in total score compared to T0 (61.15). Compared to the reference cut-off, the score is below the clinical reference range (Figure 2). In the DERS, the result obtained at T0 shows a high score (T0 = 88) that falls within the clinical reference range (cut-off: score totals clinical range 80–127). In the reassessment at T1, the value drops to 69.85 (Figure 3), showing a significant reduction (<0.01) below the cut-off. In particular, statistically significant improvement is shown in the variables “Strategies”, “Awareness”, and “Clarity” as well as in the total score (Table 5).

In summary, ten months later, at the end of the team-building, the re-evaluated nursing staff (N = 13) showed significant improvement in all variables analyzed (Table 6; Figure 4).

## 4. Discussion

Work-related stress is a common condition in many work settings and particularly within the health professions. Over the years, numerous studies have highlighted the negative effects of this form of stress on individuals, organizations, and clients/users [24].

The field of pediatric oncology, in addition to stressors affecting all categories of workers, seems to expose staff more to the onset of occupational disorders such as burnout or other medical conditions [25,26,27,28,29,30]. Some studies have recognized that staff working in this field need support to avoid burnout and the onset of other mental disorders [27,28].

Healthcare professionals in pediatric oncology face emotionally intense relationships with patients, which can lead to significant psychological stress. Factors contributing to burnout include the perception of cancer as “incurable”, the emotional strain of patient interactions, managing pain and suffering, and frequent exposure to death. Burnout negatively impacts both the well-being of healthcare workers and the quality of care they provide, as well as their relationships with patients and families.

Consistent with the literature in the field, our study found a risk of burnout, alexithymia, and emotional dysregulation among pediatric oncology nurses at baseline. An analysis of burnout level scores, in particular, showed a medium risk of personal accomplishment, a high risk of depersonalization, and a medium risk of emotional exhaustion. Similarly, alexithymia and emotional dysregulation showed clinically significant scores for these two scales. This analysis is confirmed by a recent systematic review points out that burnout is an increasingly common condition among healthcare workers in many different countries [13,14,15,16,17]; in particular, this has become more pronounced in recent years due to changes in the healthcare system during and after the COVID-19 pandemic.

Baseline scores in our study emphasize the need for effective burnout prevention strategies and treatment plans for healthcare workers.

In relation to interventions to reduce and prevent the risk of burnout, some studies have highlighted the importance of developing wellness programs for oncology staff through training, on-site counselors, mindfulness sessions, debriefing, healthy lifestyle promotion, sports promotion, emotional support, and communication management [30,31,32,33].

Increasing scientific interest in the use of art therapy in the management of burnout has led to further studies evaluating its effectiveness in preventing and reducing its risk [7]. In accordance with the literature, our study, the Art-Out pilot project, aimed to evaluate the effectiveness of team building associated with the clay workshop and to test to what extent art can be a means of reducing and preventing burnout. At the end of the Art-Out course, the nursing team was evaluated and noted a clear improvement in all the variables investigated. All analyzed scales improved in comparison to the baseline and fell significantly below the clinical reference range.

In addition to the Art-Out Project, we are actively exploring and organizing various initiatives aimed at reducing burnout and enhancing overall well-being. These activities include opportunities for team members to participate in boat outings, and attend cultural events such as film festivals, music performances, and museum visits, among others. These initiatives are designed to promote relaxation, foster social connections, and provide creative outlets to manage stress and prevent burnout.

Moreover, we are continuously assessing additional strategies to reduce burnout, including the possibility of increasing staff recruitment and reassessing work schedules to support a better work-life balance.

The limitation of our study is the sample size, which consists of only 13 participants. The refusal of some professionals (N = 7) due to time constraints further reduced the sample size, potentially impacting the representativeness of our results. Given this limitation, the therapeutic value of other team-building interventions incorporating clay therapy could be investigated in other oncology and pediatric oncology settings. Additionally, the sample is limited to nurses not involving other professionals who are also at risk of burnout. To address this gap, phase two of the project is currently underway to involve medical staff in the intervention.

## 5. Conclusions

Overall, our results highlight the emotional burden of pediatric oncology healthcare workers and the need for structured support programs to enhance the staff’s well-being and clinical care quality.

The existing literature highlights how art can be an effective tool to improve the well-being of healthcare personnel. Based on this evidence, the pilot project Art-Out was initiated specifically incorporating clay therapy. Team building empowered by expressive work resulted in reduced levels of burnout and improved analyzed variables. The results of the study showed a reduction in burnout and a marked improvement in nurses’ awareness and acceptance of their emotions as well as the acquisition of appropriate emotional regulation strategies. Our analysis shows that this approach promoted well-being and improved the emotional climate of the nursing staff.

The group course was positively received by the participants and allowed a better understanding of the emotional challenges of the nursing team and how to improve the emotional management of patients’ physical and psychological suffering. The shared group experience enhanced the creative process and promoted the psychological well-being of the nursing staff.

In conclusion, the results of this pilot study highlight the significant emotional burden experienced by healthcare workers, showing the importance of implementing innovative supportive interventions to prevent burnout. These interventions have the potential to enhance workers’ well-being, improve team dynamics, and contribute to the quality of care provided to patients.

## Figures and Tables

**Figure 1 cancers-17-01099-f001:**
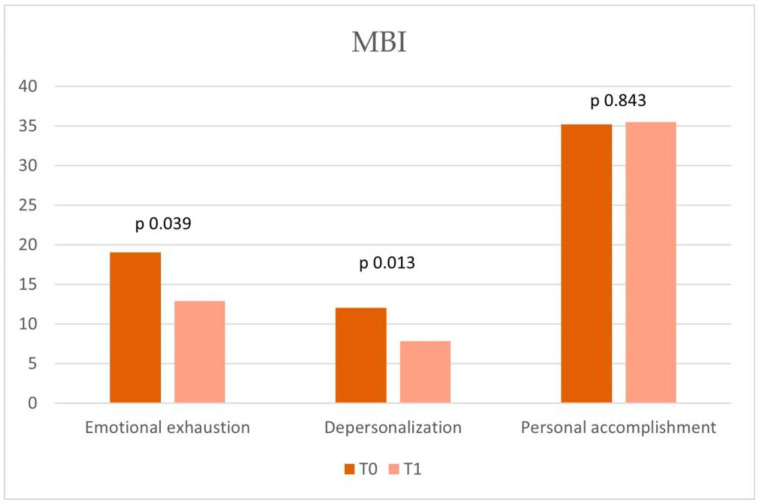
Maslach Burnout Inventory (MBI): Comparative analysis of T0 and T1 scores across the three burnout dimensions.

**Figure 2 cancers-17-01099-f002:**
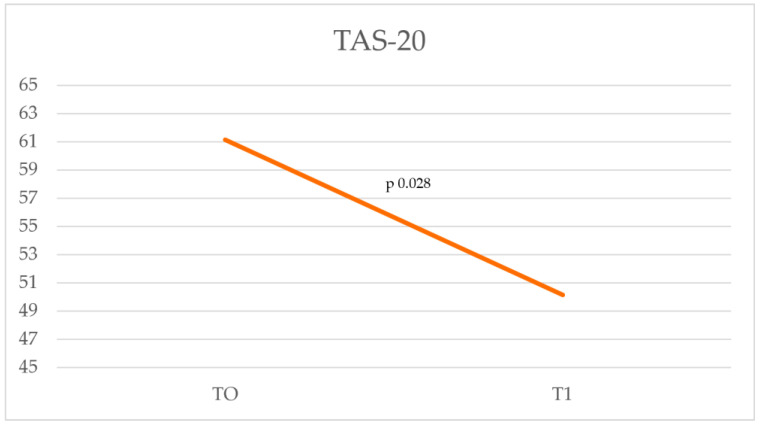
Comparison of T0 and T1 scores on the Toronto Alexithymia Scale (TAS-20), assessing changes in alexithymia levels over time.

**Figure 3 cancers-17-01099-f003:**
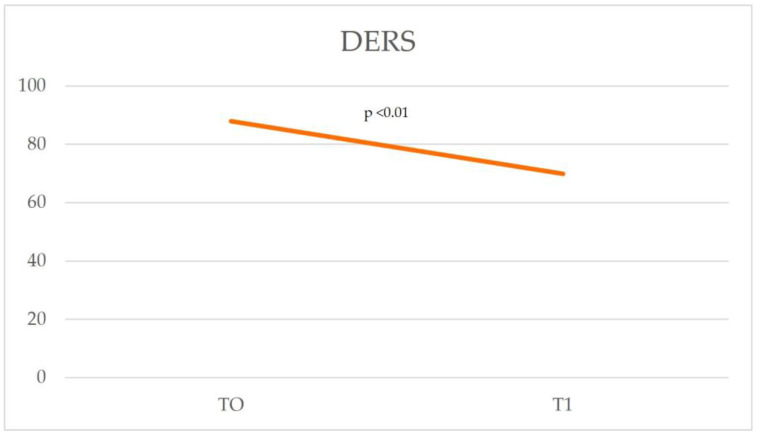
Comparison of T0 and T1 scores on the Difficulties in Emotion Regulation Scale (DERS), illustrating changes in emotion regulation difficulties over time.

**Figure 4 cancers-17-01099-f004:**
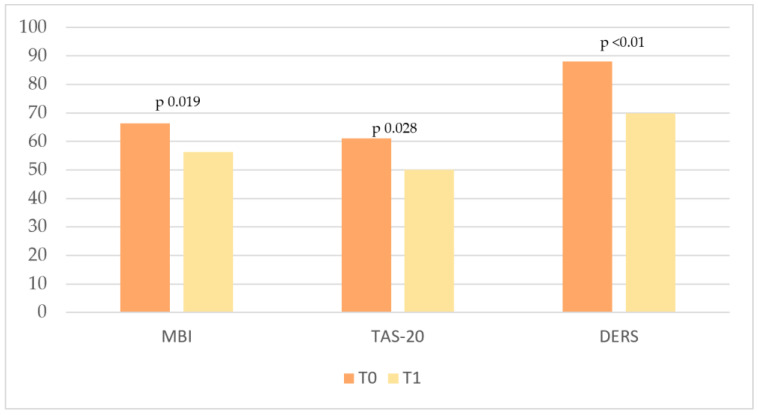
T0 and T1 scores for MBI, TAS, and DERS (Total Score), showing changes over time.

**Table 1 cancers-17-01099-t001:** Demographic variables (N = 13).

Demographic Variables	N (%)	Means ± SD
Age, years		43.31
Gender		
Male	1 (7.7)	
Female	12 (92.3)	
Role in pediatric oncology		
Nurse	13 (100)	
Professional seniority		
0–10	5 (38.5)	
11–20	0	
>20	8 (38.5)	
Professional seniority in pediatric oncology		
0–10	6 (46.1)	
11–20	4 (30.8)	
>20	3 (23.1)	
Psychotherapy		
Yes, still in progress	1 (7.7)	
Yes, in the past	3 (23.1)	
No, never	9 (69.2)	

**Table 2 cancers-17-01099-t002:** Maslach Burnout Inventory (MBI).

Factors	Baseline Means (SD)	Follow-Up Means (SD)	*p*-Value
Emotional exhaustion	19.077 (8.827)	12.92 (3.61)	0.039
Depersonalization	12.077 (4.132)	7.846 (3.555)	0.013
Personal accomplishment	35.231 (1.964)	35.53 (5.12)	0.843
Total score MBI	66.8 (11.41)	56.296 (7.2)	0.019

**Table 3 cancers-17-01099-t003:** Cut-off Maslach Burnout Inventory (MBI).

MBI Scale Test	Cut-Off
Emotional exhaustion	High risk ≥ 30 Medium risk 18–29 Low risk ≤ 17
Depersonalization	High risk ≥ 12 Medium risk 6–11 Low risk ≤ 5
Personal accomplishment	High risk ≤ 33 Medium risk 34–39 Low risk ≥ 40

**Table 4 cancers-17-01099-t004:** Toronto Alexithymia Scale (TAS 20).

	Baseline Means (SD)	Follow-Up Means (SD)	*p*-Value
Total score TAS 20	61.15 (8.93)	50.15 (11.82)	0.028

**Table 5 cancers-17-01099-t005:** Difficulties in Emotion Regulation Scale (DERS).

	Baseline Means (SD)	Follow-Up Means (SD)	*p*-Value
Non-Acceptance	13.923 (4.555)	12.385 (3.453)	0.616
Goals	12.846 (4.543)	11.538 (2.696)	0.714
Strategies	22.923 (2.290)	20.231 (2.351)	0.009
Impulse	11.385 (3.798)	10.538 (2.602)	0.760
Awareness	13.231 (1.235)	9.231 (3.492)	0.001
Clarity	13.692 (1.601)	5.923 (2.397)	<0.001
Total score DERS	88 (9.10)	69.85 (10.04)	<0.01

**Table 6 cancers-17-01099-t006:** T0 and T1 of all variables analyzed.

	Baseline Means (SD)	Follow-Up Means (SD)	*p*-Value
MBI	66.38 (11.41)	56.296 (7.2)	0.019
TAS-20	61.15 (8.93)	50.15 (11.82)	0.028
DERS	88 (9.10)	69.85 (10.04)	<0.01

## Data Availability

More detailed data can be obtained by contacting the author of the communication.

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
