# Peer review of "Burnout in Pediatric Oncology: Team Building and Clay Therapy as a Strategy to Improve Emotional Climate and Group Dynamics in a Nursing Staff"

_cancers, 2025, doi:10.3390/cancers17071099_

Round 1

Reviewer 1 Report

Comments and Suggestions for Authors

Burn out in the oncology field is really high and the authors designed this interesting study which is novel to deal with stress in the field. The study is small but sound design and result are encouraging. 

Author Response

REVIEWER 1

Burn out in the oncology field is really high and the authors designed this interesting study which is novel to deal with stress in the field. The study is small but sound design and result are encouraging.

R. Thank you for your thoughtful comment. We appreciate your recognition of the novelty and importance of this study in addressing stress in the oncology field. We also value your positive feedback on the study design and results, and we hope it will contribute to further research and improvements in this area.

Reviewer 2 Report

Comments and Suggestions for Authors

Thank you for having addressed an interesting and relevant topic, such as the nursing staff burnout in a pediatric oncology unit.

Title:

You should consider adding “in a nursing staff”. This would be useful to identify the project participants.

Abstract:

Line 32 and 34: In the previous lines you specify that burnout levels were assessed before (T0) and after (T1) the team-building course, however, you do not specify at which time points alexithymia and emotional regulation difficulties were assessed.

Line 38: “health care workers” in a pediatric oncology unit should also be physiotherapists, dieticians and others, therefore I suggest you consider using the term “nurses” rather than “health care workers”.

  1. Introduction:

Page 2, line 51: I guess the capital letter of “Physical” is a typing error.

Page 2, line 68: I think you wanted to say “(…) healthcare workers”.

  1. Materials and methods:

Page 3, lines 91-92: it seems that all these assessment tools evaluate the same outcome (burnout), however, they consider three different aspects and only the Maslach Burnout Inventory assess burnout. You should consider rewording this sentence.

Page 3, line 94: here you refer to “two subgroups”, however, you name this methodological aspect only on line 100. It is therefore difficult for the reader to understand here what you are referring to.

2.1. Participants

 In this section you give some data that is expected to be found in the results section. You give the number of participants (lines 98-99) and then you describe their characteristics (lines 105-108). You should consider moving this data to the results section.

2.2. Tools

It would be nice to describe the different assessment tools characteristics in a consistent way. To achieve this goal, you should add to the description of Toronto Alexithymia Scale how much it is used in literature and move the sentence that refers to the meaning of a score of 61 or higher at the end of the period. You should also add to the Difficulties in Emotion Regulation Scale description how much it is used in literature, the number of items of each sub-scale and the scoring meaning.

2.3. The Art-Out Project

Page 5, line 171: you should better explain how was organized both the brainstorming sessions and the focus groups are organized in terms of number of participants, number of sessions/focus groups, and duration.

Page 5, line 182: you repeated twice the word “listening”.

  1. Results:

Page 5, line 190: in the Materials and Methods section you describe the three assessment tools stating from the Maslach Burnout Inventory and finishing with the Difficulties in Emotion Regulation Scale. It would be good to maintain the same order each time you list these in the text.

Page 5, line 191, 199 and 205: you have already explained these acronyms in the previous parts of the text, it is not necessary to repeat it here.

  1. Discussion:

Data is generally not reported in the discussion section. Data should only be reported in the results section, while in the discussion section it is expected that authors will summarize key results with reference to study objectives and interpretate their results considering other relevant evidence on the topic. I suggest you rewording this section taking into consideration the STROBE Checklist for cross-sectional studies.  

  1. Conclusions:

Page 8, line 259: you should not report any reference in the conclusions section.

Page 8, line 260: Be careful with the terminology, in the other parts of the text you name the project without using the quotation marks.

The strengths and limitations of the study should be moved to the discussion section.

Author Response

REVIEWER 2

Thank you for having addressed an interesting and relevant topic, such as the nursing staff burnout in a pediatric oncology unit.

Title:

You should consider adding “in a nursing staff”. This would be useful to identify the project participants.

R. Thank you for highlighting this. We agree with your suggestion and have updated the title to specify that the study involves the nursing staff (line 4).

Abstract:

Line 32 and 34: In the previous lines you specify that burnout levels were assessed before (T0) and after (T1) the team-building course, however, you do not specify at which time points alexithymia and emotional regulation difficulties were assessed

R. Thank you for your comment. We have revised the abstract to specify that all three tests were administered at T0 and then at T1 (lines 31-32).

Line 38: “health care workers” in a pediatric oncology unit should also be physiotherapists, dieticians and others, therefore I suggest you consider using the term “nurses” rather than “health care workers”

R. Thank you for pointing this out. We have made the change and used the term "nurses" instead of "health care workers" (line 38).

  1. Introduction:

Page 2, line 51: I guess the capital letter of “Physical” is a typing error.

R. We have changed and corrected “physical” ( typing error)(line 51)

Page 2, line 68: I think you wanted to say “(…) healthcare workers”

R.Yes, we intented to say “(…) healthcare workers” (line 68)

2. Materials and methods:

Page 3, lines 91-92: it seems that all these assessment tools evaluate the same outcome (burnout), however, they consider three different aspects and only the Maslach Burnout Inventory assess burnout. You should consider rewording this sentence.

R.Thank you for pointing this out. We have reworded this sentence (lines 90-91).

Page 3, line 94: here you refer to “two subgroups”, however, you name this methodological aspect only on line 100. It is therefore difficult for the reader to understand here what you are referring to.

R. We have revised the sentence and removed the term "two subgroups" as it was confusing. In fact, the group of nurses involved in the study is a single group. We were referring to the two shifts of the same team-building meeting, which were organized to allow all participants to take part, regardless of their work shift (lines 97-98).

2.1. Participants

 In this section you give some data that is expected to be found in the results section. You give the number of participants (lines 98-99) and then you describe their characteristics (lines 105-108). You should consider moving this data to the results section.

R. We have relocated the participant data to the Results section (lines 193-203).

2.2. Tools

It would be nice to describe the different assessment tools characteristics in a consistent way. To achieve this goal, you should add to the description of Toronto Alexithymia Scale how much it is used in literature and move the sentence that refers to the meaning of a score of 61 or higher at the end of the period. You should also add to the Difficulties in Emotion Regulation Scale description how much it is used in literature, the number of items of each sub-scale and the scoring meaning.

R. We have included the literature related to the two scales (DERS and TAS 20) and edited and supplemented the text as suggested (lines 128-132, 138-142, 143-152)

2.3. The Art-Out Project

Page 5, line 171: you should better explain how was organized both the brainstorming sessions and the focus groups are organized in terms of number of participants, number of sessions/focus groups, and duration.

R. Thank you for pointing this out. We have edited and supplemented the text as suggested (lines 159-164, 172-181).

Page 5, line 182: you repeated twice the word “listening”.

R. We have corrected the word “listening”.

3. Results:

Page 5, line 190: in the Materials and Methods section you describe the three assessment tools stating from the Maslach Burnout Inventory and finishing with the Difficulties in Emotion Regulation Scale. It would be good to maintain the same order each time you list these in the text.

R. We have modified as suggested (lines 104-205).

Page 5, line 191, 199 and 205: you have already explained these acronyms in the previous parts of the text, it is not necessary to repeat it here.

R. Acronyms: We have modified the text as suggested

4. Discussion:

Data is generally not reported in the discussion section. Data should only be reported in the results section, while in the discussion section it is expected that authors will summarize key results with reference to study objectives and interpretate their results considering other relevant evidence on the topic. I suggest you rewording this section taking into consideration the STROBE Checklist for cross-sectional studies. 

R. Thank you for your helpful suggestion. We have modified the sections as recommended, ensuring that data is now reported only in the results section. In the discussion, we have focused on summarizing the key results in relation to the study objectives and interpreting them in the context of relevant evidence (lines 252-270).

5. Conclusions:

Page 8, line 259: you should not report any reference in the conclusions section.

R. We have revised the text by removing the references.

Page 8, line 260: Be careful with the terminology, in the other parts of the text you name the project without using the quotation marks.

R. We have corrected the text by removing the quotation marks as suggested.

The strengths and limitations of the study should be moved to the discussion section.

R. We have moved the limitations of the study to the Discussion section (lines 271-278).

Reviewer 3 Report

Comments and Suggestions for Authors

Burnout among the healthcare workers is an emergent problem in many developed and developed countries worldwide. Detecting the burnout of the medical teams is essential for early intervention and guarantee the quality of medical management in patients. In this manuscript, the authors study the possible burnout of the nursery stuff in a Pediatric Oncology Department in Italy. The report reminds to the medical community to interfere early to protect the healthcare providers against the burnout and suggests a tool to resolve it. However, the study has to be presented in better clarity and present stronger their conclusions. Thus, I accept this manuscript for publication after major corrections. Herein are some comments for the authors:

  1. The authors must assess and describe the burnout of the medical doctors in the whole of hierarchy (consultants, fellows, and assistants) in the same department. If the nurse stuff is healthy, but the medical stuff cannot be functioned properly, the optimal management in the pediatric oncology patients is compromised.
  1. The section of Art-Out Project must be presented before the section of statistical analysis.
  2. In the first sentence of Art-Out Project section (line 166), the authors have to explain what it is about. The readers understand much later that it is about a meeting with the nurse team members.
  3. Is the Art-Out Project enough to manage the stuff burnout? Are there any additional measures that have to be implied, e.g. increasing the stuff hiring, reducing the week working hours, etc.
  4. In the figures, statistical significance before and after the implication of “Art-Out Project” has to be noted (if it exists)
  5. In the discussion part, the authors have to structure paragraphs.
  6. In the discussion, the authors have to hypothesize or explain the reasons that the nursing stuff was exhausted
  7. There is no paragraph with only one sentence. Please, restructure the manuscript when there is one sentence as paragraph, e.g. lines 75-76, lines 207-210, lines 211-213.

Author Response

REVIEWER 3

Burnout among the healthcare workers is an emergent problem in many developed and developed countries worldwide. Detecting the burnout of the medical teams is essential for early intervention and guarantee the quality of medical management in patients. In this manuscript, the authors study the possible burnout of the nursery stuff in a Pediatric Oncology Department in Italy. The report reminds to the medical community to interfere early to protect the healthcare providers against the burnout and suggests a tool to resolve it. However, the study has to be presented in better clarity and present stronger their conclusions. Thus, I accept this manuscript for publication after major corrections. Herein are some comments for the authors:

1. The authors must assess and describe the burnout of the medical doctors in the whole of hierarchy (consultants, fellows, and assistants) in the same department. If the nurse stuff is healthy, but the medical stuff cannot be functioned properly, the optimal management in the pediatric oncology patients is compromised.

R. Thank you for your request regarding the assessment of burnout among medical doctors across the entire hierarchy (consultants, fellows, and assistants) in the department. I would like to inform you that a similar project is currently underway to assess burnout among medical doctors as part of a broader initiative to evaluate the overall level of burnout in healthcare professionals. The two projects—one focusing on nurses and the other on medical doctors—have been kept separate due to the distinct roles, responsibilities, operational activities, and tasks between the two groups. This approach allows for a more accurate and tailored evaluation of each group's specific needs and challenges in relation to burnout. We fully understand that if the medical staff is unable to function optimally, despite the well-being of the nursing team, the overall management of pediatric oncology patients may be compromised. Therefore, both projects are essential in ensuring the best possible care and well-being for both the patients and healthcare professionals involved.

2. The section of Art-Out Project must be presented before the section of statistical analysis

R. Thank you for pointing this out. We have moved the section Art-Out Project as suggested

3. In the first sentence of Art-Out Project section (line 166), the authors have to explain what it is about. The readers understand much later that it is about a meeting with the nurse team members.

R. Thank you for your suggestion. We agree and have made the changes as recommended (lines 156-157).

4. Is the Art-Out Project enough to manage the stuff burnout? Are there any additional measures that have to be implied, e.g. increasing the stuff hiring, reducing the week working hours, etc.

R. Thank you for your question regarding the Art-Out Project and its effectiveness in managing staff burnout. While the Art-Out Project is an important initiative, it is just one of several measures currently being implemented to address burnout within the team. In addition to the Art-Out Project, we are also exploring and organizing other activities aimed at reducing burnout and improving overall well-being. These initiatives include opportunities for team members to participate in boat outings, attend cultural events such as film festivals, music performances, and museum visits, among others. These activities are designed to offer relaxation, foster social connections, and provide creative outlets to help manage stress and prevent burnout. Furthermore, we continue to evaluate additional strategies that may contribute to reducing burnout, including exploring the possibility of increasing staff hiring and reassessing work schedules to allow for a better work-life balance. We are committed to implementing a comprehensive approach to staff well-being and will continue to consider and introduce new measures as necessary.

5. In the figures, statistical significance before and after the implication of “Art-Out Project” has to be noted (if it exists)

R. The p-value has been reported in the table.

6. In the discussion part, the authors have to structure paragraphs.

R. Thank you for your feedback regarding the structure of the discussion section. The Discussion section has been reviewed.

7. In the discussion, the authors have to hypothesize or explain the reasons that the nursing stuff was exhausted

R. The Discussion has been revised to include some explanations about burnout in the nursing staff.

8. There is no paragraph with only one sentence. Please, restructure the manuscript when there is one sentence as paragraph, e.g. lines 75-76, lines 207-210, lines 211-213.

R. Thank you for your comment. We agree with your suggestion and have made the necessary modifications

Round 2

Reviewer 2 Report

Comments and Suggestions for Authors

Dear authors, I believe that all the modifications you made improved the study reporting.

Author Response

Dear authors, I believe that all the modifications you made improved the study reporting

R. Thank you for your comment. We greatly appreciate your feedback and are glad to hear that the modifications have enhanced the study reporting.

Reviewer 3 Report

Comments and Suggestions for Authors

After the integration of modifications, the manuscript is much better presented and more comprehensive. However, there are some additional corrections that I suggest to the authors

  1. Lines 42. The authors have to create paragraphs in the induction and in the conclusions.
  2. Lines 237 It is not clear the correspondence between figures and their legends
  3. In the figure 2, which is the statistical analysis that the p value 0,019 corresponds?
  4. In the discussion part, the authors have to explore any additional measures, expect the Art-Out Project, that can help to reduce the burn-out (it has already been asked in my previous report).  Additional studies that describe the burnout of medical stuff have to be reported and be commented.

Author Response

We thank the reviewer for their comments. Revisions have been made to the text. All changes are highlighted in green.

1. Lines 42. The authors have to create paragraphs in the induction and in the conclusions. 

 R. Paragraphs in the induction and in the conclusions have been added

2. Lines 237 It is not clear the correspondence between figures and their legends

R. The legends have been rewritten to provide clearer clarification for the reader

3. In the figure 2, which is the statistical analysis that the p value 0,019 corresponds?

R. Figure 2 has been revised, and p-values have been added for each evaluated parameter.

4. In the discussion part, the authors have to explore any additional measures, expect the Art-Out Project, that can help to reduce the burn-out (it has already been asked in my previous report).  Additional studies that describe the burnout of medical stuff have to be reported and be commented.

R. Based on your feedback, we have added additional data and explored further measures, beyond the Art-Out Project, that can help reduce burnout. Additionally, we have included relevant studies describing burnout. Some references have been added